# Metoclopramide and Levosulpiride Use and Subsequent Levodopa Prescription in the Korean Elderly: The Prescribing Cascade

**DOI:** 10.3390/jcm8091496

**Published:** 2019-09-19

**Authors:** Youn Huh, Do-Hoon Kim, Moonyoung Choi, Joo-Hyun Park, Do-Young Kwon, Jin-Hyung Jung, Kyungdo Han, Yong-Gyu Park

**Affiliations:** 1Department of Family Medicine, Inje University Ilsan Paik Hospital, College of Medicine, Inje University, 170, Juhwa-ro, Ilsanseo-gu, Goyang-si, Gyeonggi-do 10380, Korea; higjdus1@naver.com; 2Department of Family Medicine, Korea University Ansan Hospital, Korea University College of Medicine, 123, Jeokgeum-ro, Danwon-gu, Ansan-si, Gyeonggi-do 15355, Korea; joohyun_park@naver.com; 3Department of Neurology, Korea University Ansan Hospital, Korea University College of Medicine, 123, Jeokgeum-ro, Danwon-gu, Ansan-si, Gyeonggi-do 15355, Korea; kwondoya@hanmail.net; 4Department of Biostatistics, College of Medicine, The Catholic University of Korea, Seoul 03083, Korea; jungjin115@naver.com (J.-H.J.); hkd917@naver.com (K.H.); ygpark@catholic.ac.kr (Y.-G.P.)

**Keywords:** elderly, Parkinson’s disease, prescribing cascade, metoclopramide, levosulpiride, Levodopa medications, drug-induced

## Abstract

The aim of this study was to investigate the prescribing cascade phenomenon of dopaminergic drugs such as levodopa in the management of gastroprokinetic drugs induced parkinsonism. Based on the Korea National Health Insurance Service (NHIS)-Senior Cohort Database, we analyzed patients aged ≥65 years, between 2009 and 2013, who obtained new prescriptions for levodopa through the NHIS during this period. Those who were prescribed levodopa from 2002 to 2008 were excluded, only patients who were prescribed metoclopramide and levosulpiride within 90 days of receiving the levodopa prescription were included. Those who did not receive levodopa were used as a control group for 1:3 age and sex matching. We assessed 1824 and 1197 levodopa cases for metoclopramide and levosulpiride use, respectively. The matched controls for each levodopa case were 5472 and 3591, respectively. We used conditional logistic regression to determine the odds ratio (OR) for initiation of levodopa therapy in patients using metoclopramide and levosulpiride, relative to nonusers, after adjusting for age, sex, and exposure to antipsychotic medication. Both metoclopramide (OR = 3.04; 95% confidence interval, CI, 2.46–3.77) and levosulpiride (OR = 3.32; 95% CI, 2.56–4.3) users were three times more likely to begin using medication containing levodopa, compared to nonusers. Metoclopramide and levosulpiride were frequently prescribed within 90 days of receiving a prescription for levodopa. Before prescribing levodopa, it should be considered whether the adverse event is actually a side effect caused by metoclopramide and levosulpiride.

## 1. Introduction

Adverse events associated with medication are quite common. And these place a significant burden on the healthcare system in terms of both health outcomes and costs. A prescribing cascade occurs when a new medication is prescribed to “treat” an adverse drug reaction associated with another medication, based on the misdiagnosis that a new medical condition requiring treatment is present [1]. Prescribing cascades resulting from both recognized and unrecognized adverse reactions may be hazardous to patients. Some studies reported that older adults may be at a higher risk of prescribing cascades than younger adults [2,3]. This may be because they are more likely to suffer from an adverse drug reaction than the younger population, resulting in a higher possibility of misinterpretation of the reaction as the onset of a new medical condition [1]. 

The Diagnostic and Statistical Manual of Mental Disorders, Fifth Edition (DSM-V), [4] defines drug-induced parkinsonism (DIP) as the presence of resting tremor, muscular rigidity, akinesia, or bradykinesia, developing within a few weeks of starting or raising the dosage of a medication (typically a neuroleptic) or after reducing the dosage of an antiparkinsonian agent. Drug-induced movement disorders include DIP, tardive dyskinesia (TD), tardive dystonia, akathisia, myoclonus, and tremors. Amongst these, DIP is the most common movement disorder induced by medications that block dopamine receptors. DIP is the second-most common cause of parkinsonism, and a major cause of the misleading diagnosis of Parkinson’s disease (PD). It was initially considered to be present in approximately 4–40% of patients treated with the first neuroleptics [5], however, it has been reported as a complication of treatment.

The risk factors of DIP may be related to several variables including age [6,7], female gender [6], length of exposure [8], dose, and treatment with multiple drugs. Age is the most obvious risk factor for DIP [9,10,11], because dopamine concentrations decrease and nigral cells degenerate with age [12]. 

Medications that cause drug-induced parkinsonism include typical and atypical antipsychotics, dopamine depleters, antiemetics, and calcium channel blockers [13,14,15]. Gastroprokinetic drugs are potent inducers of DIP, especially according to reports in Korea [16]. Metoclopramide and levosulpiride are well-known antiemetic and gastroprokinetic agents commonly used for the treatment of nausea, vomiting, gastroparesis, gastroesophageal reflux disease, and migraine [17,18,19,20], and are gastrointestinal prokinetic medications mediated through a blockade of their enteric inhibitory D2 receptors [21]. Besides binding to receptors in the peripheral end organs, thus inducing antiemetic effects via D2 receptor blockade in the area postrema, they also antagonize central D2 receptors, leading to adverse effects including hyperprolactinemia and extrapyramidal side effects. All prokinetics with D2-receptor-antagonizing properties have been found to induce extrapyramidal side effects, although the extent of symptoms varies [13].

Since older age is a risk factor for DIP, the present study aimed to determine the association between the initiation of levodopa therapy and the use of metoclopramide or levosulpiride in the older population. For this purpose, we analyzed the National Health Insurance Service (NHIS)-Senior Cohort Database.

## 2. Materials and Methods

### 2.1. Setting and Participants

The NHIS-Senior Cohort Database consists of a randomly selected 10% of the Korean population aged ≥60 years (550,000 persons), since the end of 2002. The database contains information including demographics, income (including socioeconomic factors), medical records, health screening records, and responses to a questionnaire. In Korea, the National Health Insurance organization recommends health screening in all adults, every two years.

### 2.2. Patient Selection and Criteria

This study was conducted using two antiemetics: metoclopramide and levosulpiride. For cohorts, we identified patients aged ≥65 years, between 2009 and 2013, who obtained new prescriptions for levodopa through the NHIS during this period. We extracted the control group and the case group by the method shown in Figure 1.

### 2.3. Medication Exposure

Based on the Korea National Health Insurance Service (NHIS)-Senior Cohort Database, we analyzed patients aged ≥65 years, between 2009 and 2013, who obtained new prescriptions for levodopa through the NHIS during the same time period. Those who were prescribed levodopa from 2002 to 2008 were excluded, only patients who were prescribed metoclopramide and levosulpiride within 90 days of receiving the levodopa prescription were included, and those who did not receive levodopa were used as a control group for 1:3 age- and sex matching. We assessed 1824 and 1197 levodopa cases for metoclopramide and levosulpiride use, respectively; the matched controls for each levodopa case were 5472 and 3591, respectively. In addition, the prescription rate of levodopa according to the duration of metoclopramide and levosulpiride use was compared between groups. 

### 2.4. Statistical Analysis

Odds ratios (ORs) were calculated to assess the association between antiemetic use and initiation of levodopa. All statistical analyses were performed using SAS version 9.3 software for conditional logistic regression, which considered the initiation of levodopa therapy as the outcome of interest. Additionally, obtaining a prescription for metoclopramide or levosulpiride in the last 90 days prior to the index date was considered the exposure of interest. Then, we developed a regression model that adjusted for age, sex, nursing home resident status, history of hospital care in the last 120 days prior to the index date, and any use of antipsychotic medication (such as chlorpromazine, perphenazine, haloperidol, pimozide, sulpiride, risperidone, olanzapine, ziprasidone, and aripiprazole) during the 90-day period prior to the index date. After initial analyses, interaction terms were added to the model to determine whether the effect of metoclopramide or levosulpiride was modified by age, sex, or exposure to antipsychotic medication. All *P*-values were two-tailed; a *P*-value of < 0.05 indicated statistical significance.

## 3. Results

### 3.1. Metoclopramide Use and Subsequent Levodopa Prescription 

Table 1 displays the characteristics of the study population for metoclopramide, which included 1824 cases and 5472 controls. The study population was analyzed by sex, age, use or non-use of metoclopramide, and duration of metoclopramide use. Those who were prescribed metoclopramide were about three times more likely to be prescribed levodopa than those who were not (OR = 3.04; 95% confidence interval, CI, 2.46–3.77) (Table 2). This increased risk persisted even after adjusting for age, sex, and use of antipsychotics (OR = 2.94; 95% CI, 2.35–3.67). As the duration of metoclopramide use increased, the likelihood of taking levodopa increased as well; OR was 2.93 (95% CI, 2.34–3.68) for the 1–19 days group, and 4.18 (95% CI, 2.21–7.89) for the >20 day group.

### 3.2. Levosulpiride Use and Subsequent Levodopa Prescription 

The study population for levosulpiride consisted of 1197 cases and 3591 controls and they were also analyzed in terms of sex, age, use or non-use of levosulpiride, and the duration of use (Table 3). Table 4 exhibits the risk of using levodopa, according to pre-existing use or non-use of levosulpiride. Those individuals who were prescribed levosulpiride were around three times more likely to be prescribed levodopa (OR = 3.32; 95% CI, 2.56–4.30). After adjusting for age, sex, and use of antipsychotics, the risk increased slightly (OR = 3.30; 95% CI, 2.52–4.32). As the duration of levosulpiride use was longer, the likelihood of taking levodopa was found to be greater; ORs were 2.29 (95% CI, 2.69–3.11) for the 1–19 days group, and 9.79 (95% CI, 5.65–16.97) for the >20 day group. 

## 4. Discussion

DIP is important because it is a common etiology of parkinsonism and is frequently either unrecognized or misdiagnosed as PD. In addition, parkinsonism in patients with DIP can be severe enough to affect daily activities and may persist for long periods of time even after terminating the consumption of the medication. Several long-term follow-up studies have revealed that 20–25% of patients with DIP develop persistent and progressive parkinsonian deficits in spite of drug discontinuation [22,23].

Many studies have already demonstrated that antiemetics are associated with DIP [16,24]. In one study in the United States, metoclopramide users were three times more likely to start using levodopa-containing drugs than nonusers (OR = 3.09; 95% CI, 2.25–4.26). In addition, increased risk with increasing daily metoclopramide dose: OR = 1.19 (95% CI, 0.50–2.81) above 0–10 mg per day, 3.33 (95% CI, 1.98–5.58) above 10–20 mg per day, and 5.25 (95% CI, 1.16–8.50) for more than 20 mg a day [16]. In our study, it was found that levodopa prescription increased when antiemetics were used during the preceding period. In clinical practice, levodopa is often prescribed to treat Parkinsonism associated with antiemetics in older adults. This may lead to polypharmacy in such patients, which causes adverse drug reactions and has a significant impact on health and cost. 

According to the results of a study on the status of drug use in the older population in 2014, 60.3% of Korean people aged ≥65 years take more than three types of medication. Given that older adults take multiple medications, it is difficult to determine if the new symptoms are a result of either adverse drug reactions, or aging, therefore, it is necessary to evaluate whether any new symptoms are caused by adverse drug reactions. As other studies have shown that DIP is dose-dependent, the subsequent prescription of levodopa in this study has also increased with longer use of gastroprokinetics [24,25]. In one such study in South Korea, ORs tended to be higher in those exposed to higher chronic cumulative doses and had significant association with increased risk of parkinsonism, depending on cumulative dose [24]. In our study, the longer the exposure period, the greater the use of levosulpiride than metoclopramide. 

In the case of metoclopramide, levodopa was subsequently used 2.82 times for the 1–19 days and 4.14 times for more than 20 days. For levosulpiridie, 2.24 times and 10.24 times levodopa prescription were followed-up, respectively. Therefore, when prescribing drugs in the elderly, they should refrain from long-term medication if not needed.

This study has the following limitations. First, not all cases in this paper can be called DIP. Because this study did not consider the NHIS Parkinson’s disease diagnostic code and analyzed people who were prescribed levodopa medication, there is a limitation that those prescribed levodopa may not necessarily be diagnosed with DIP. Second, antiparkinsonian medications other than levodopa were not considered. Because DIP treatment does not necessarily use levodopa, there are limitations that other antiparkinsonian disease treatments may also be used. Third, this study is based on the NHIS-Senior Cohort Database, which consisted of elderly adults with national health checkup results, and therefore, there may be selection bias because the database is more likely to include people concerned about their health and wellbeing and adopting a healthier lifestyle than those who do not receive regular medical checkups. Fourth, in this study, the dose dependent DIP was unknown because the daily dose was not known.

Those individuals who undergo health screening at least once every two years may have a greater interest in their health and well-being compared to those who do not and may be more likely to have a healthy lifestyle, which could have created a selection bias. 

Notwithstanding these limitations, in this large-scale study, it was found that persons taking a prokinetic medication (metoclopramide or levosulpiride) were at increased risk of being subsequently prescribed levodopa, and this risk tends to increase with longer duration of prokinetic agent use.

## 5. Conclusions

Metoclopramide and levosulpiride were frequently prescribed within 90 days of a prescription for levodopa. Before prescribing levodopa, we recommend considering whether the adverse health effect is a side effect caused by metoclopramide and levosulpiride.

## Figures and Tables

**Figure 1 jcm-08-01496-f001:**
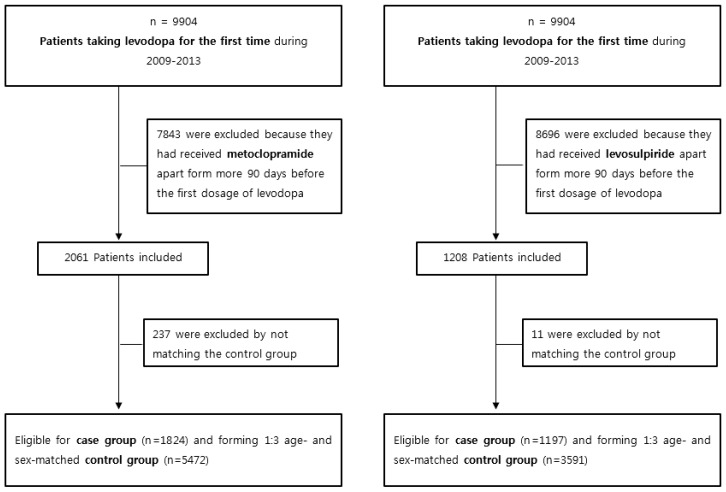
Flowchart of Study Subjects.

**Table 1 jcm-08-01496-t001:** Characteristics of study subjects in terms of preceding metoclopramide prescriptions.

		Cases *N* = 1824	Controls *N* = 5472
Sex	Male	960 (52.63)	2880 (52.63)
	Female	86 (47.37)	2592 (47.37)
Age (Years)	65–74	694 (38.05)	2082 (38.05)
	75–84	918 (50.33)	2754 (50.33)
	≥85	212 (11.62)	636 (11.62)
Exposure to Metoclopramide	No	1645 (90.19)	5280 (96.49)
	Yes	179 (9.81)	192 (3.51)
Duration of Metoclopramide prescription	None	1645 (90.19)	5280 (96.49)
	1–19 days	157 (8.61)	175 (3.2)
	≥20 days	22 (1.21)	17 (0.31)

Data are expressed as *N* (%).

**Table 2 jcm-08-01496-t002:** The odds ratios (ORs) for the initiation of levodopa, classified based on the duration of exposure to metoclopramide.

		Crude OR (95% CI)	Adjusted OR (95% CI) *
Exposure to metoclopramide	No	1 **	1 **
	Yes	3.04 (2.46,3.77)	2.94 (2.35,3.67)
Duration	None	1 **	1 **
	1–19 days	2.93 (2.34,3.68)	2.82 (2.23,3.56)
	≥20 days	4.18 (2.21,7.89)	4.14 (2.15,7.98)

Abbreviations: CI, confidence interval; OR, odds ratio. * Adjusted for age, sex, and exposure to antipsychotic medications. ** Referent category.

**Table 3 jcm-08-01496-t003:** Characteristics of study subjects in terms of preceding levosulpiride prescriptions.

		Cases *N* = 1197	Controls *N* = 3591
Sex	Male	609 (50.88)	1827 (50.88)
	Female	588 (49.12)	1764 (49.12)
Age (Years)	65–74	472 (39.43)	1416 (39.43)
	75–84	584 (48.79)	1752 (48.79)
	≥85	141 (11.78)	423 (11.78)
Exposure to levosulpiride	No	1070 (89.39)	3465 (96.49)
	Yes	127 (10.61)	126 (3.51)
Duration of levosulpiride prescription	None	1070 (89.39)	3465 (96.49)
	1–19 days	75 (6.27)	109 (3.04)
	≥20 days	52 (4.34)	17 (0.47)

Data are expressed as *N* (%).

**Table 4 jcm-08-01496-t004:** The odds ratios (ORs) for the initiation of levodopa, classified based on the duration of exposure to levosulpiride.

		Crude OR (95% CI)	Adjusted OR (95% CI) *
Exposure to Levosulpiride	No	1 **	1 **
	Yes	3.32 (2.56,4.3)	3.3 (2.52,4.32)
Duration	Non	1 **	1 **
	1–19 days	2.29 (1.69,3.11)	2.24 (1.64,3.08)
	≥20 days	9.79 (5.65,16.97)	10.24 (5.82,17.99)

Abbreviations: CI, confidence interval; OR, odds ratio. * Adjusted for age, sex, and exposure to antipsychotic medications. ** Referent category.

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
