# Peer review of "Metoclopramide and Levosulpiride Use and Subsequent Levodopa Prescription in the Korean Elderly: The Prescribing Cascade"

_jcm, 2019, doi:10.3390/jcm8091496_

Round 1

Reviewer 1 Report

Abstract:

Page 1, Line 21 – Would recommend avoiding the word ‘recruited’ for retrospective studies. Page 1, Line 26 – you noted ‘for the initiation of antiparkinson’s therapy’…is this just levodopa or any PD medications? Please clarify.

Introduction

Page 1, Line 42 – this paper found and association with age and prescribing cascade (and also polypharmacy and prescribing cascades) (PMID: 30033126) – please include. Page 2, Line 59 – please include a reference for the list of medications that can cause DIP. Page 2, Line 67 – would recommend changing to “Since older age is a risk factor for DIP, the present study…”

Methods

Page 2, Line 82 – I do not see a Figure 2? Page 3, Line 80 – how did you define new prescriptions? Did they have a certain amount of time in the data prior to their first dose, such as 12 months? This needs to be defined and adherence to – if a person first prescription is 14 days following enrollment then we don’t know if they are truly a new user

Results

Page 3, Figure 1 – In general, I had a lot of trouble following this figure. There appears to be a lot of slang on here 9904 patients taking levodopa for the first time à 9904 levodopa new-users I don’t know what ‘washed out’ means? Excluded? It seems like 75% of patients being excluded because they were previously on metoclopramide (near 90% for levosulpiridie) seems excessively high. Is this really true? 237 ‘washed out’…do you mean excluded? Page 3, Line 88 – are these newly prescribed or any fill? Please clarify. Page 4, Line 115 – shouldn’t it be 1-20 day group and not 0-20 group since ‘none’ is separate in the analysis on Table 2 (this would be the same for levosulpriride)

Discussion

Page 5, Lines 144-152 – this is doesn’t appear to add to the study since it is discussing bench science and not the clinical implications. Page 5, Line 153-158 – this seems to be more relevant in the introduction than the discussion Page 5, Line 159 – noted ‘many studies have demonstrated that antiemetics are associated with DIP” but does not cite – please cite these many studies To note, the previous study evaluating this prescribing cascade was not cite in this manuscript (PMID: 7500509) which should both be mentioned in the intro and here. Especially since your results are very similar to the results by Avorn et al (and that the number of exposure days being dose dependent is new to your study). Page 5, Line 168 – “As other studies”, please cite and described these other studies. Page 6, Line 175 – “Not all cases of this paper can be called DIP” – I’m not sure what you mean. Nonetheless, I think an explanation is needed. Page 6, 175 – “PD medications other than levodopa were not considered” Need to explain why not (levodopa is the primarily first line agent for parkinsonism?) Page 6, line 180 – I’m having trouble connecting your explanation to selection bias – isn’t it applicable to both treatment and control?

Conclusion

Typically, I don’t see citations in the conclusion I don’t think discussing the steps to prevent prescribing cascade is relevant part of your conclusion At no point do you mention metoclopramide, levosulpiride, or levodopa in your conclusion – focus on the conclusion of you results and not general prescribing cascades.

Author Response

[Point by point response to the Reviewer’s comments]

Reviewer #1’s comments:

Abstract

1) Page 1, Line 21 – Would recommend avoiding the word ‘recruited’ for retrospective studies.

[Answer] Thank you for your comments. The phrase was edited as follows: ‘we analyzed older adults’.

2) Page 1, Line 26 – you noted ‘for the initiation of antiparkinson’s therapy’…is this just levodopa or any PD medications? Please clarify.

[Answer] Thank you for your comments. We modified the word as follows: ‘for the initiation of levodopa therapy’.

Introduction
3) Page 1, Line 42 – this paper found and association with age and prescribing cascade (and also polypharmacy and prescribing cascades) (PMID: 30033126) – please include.

[Answer] Thank you for your comments. As you recommended, we added and edited the following in the ‘Introduction’ section, page 1: Some studies reported that older adults may be at a higher risk of prescribing cascades than younger adults.[1,2]

4) Page 2, Line 59 – please include a reference for the list of medications that can cause DIP.

[Answer] Thank you for your comments. We added the references for the list of medications that can cause DIP.[3-5]

5)  Page 2, Line 67 – would recommend changing to “Since older age is a risk factor for DIP, the present study…”
[Answer] Thank you for your comments. As you recommended, the phrase was edited as follows: ‘Since older age is a risk factor for DIP, the present study aimed to determine the association between the initiation of levodopa therapy and the use of metoclopramide or levosulpiride in the older population’.

Methods
6) Page 2, Line 82 – I do not see a Figure 2?

[Answer] Thank you for your comments. Figure 2 does not exist. We delete that.

7) Page 3, Line 80 – how did you define new prescriptions? Did they have a certain amount of time in the data prior to their first dose, such as 12 months? This needs to be defined and adherence to – if a person first prescription is 14 days following enrollment then we don’t know if they are truly a new user 

[Answer] Thank you for your comments. We are sorry to fail to clarify the definition of the new prescriptions. To be exact, Based on Korea National Health Insurance Survey (NHIS)-Senior Cohort Database, we analyzed patients aged ≥65 years, between 2009 and 2013, who filled new prescriptions for levodopa through NHIS during the same time period. Those who were prescribed levodopa from 2002 to 2008 were excluded, only patients who were prescribed metocloprimide and levosulpiride within 90 days receiving the levodopa prescription were included. And those who did not receive levopa were used as a control group for 1: 3 age- and sex matching. We assessed 1824 and 1197 levodopa cases for metoclopramide and levosulpiride use, respectively; the matched controls for each levodopa case were 5472 and 3591, respectively. The new definition is described in detail in Page1 Abstract and Page 3 method.

Results
8) Page 3, Figure 1 – In general, I had a lot of trouble following this figure. There appears to be a lot of slang on here 9904 patients taking levodopa for the first time à 9904 levodopa new-users I don’t know what ‘washed out’ means? Excluded? It seems like 75% of patients being excluded because they were previously on metoclopramide (near 90% for levosulpiridie) seems excessively high. Is this really true? 237 ‘washed out’…do you mean excluded?

[Answer] Thank you for your comments. In Korea, prescriptions for gastrointestinal prokinetics such as metoclopramide and levosulfiride are common. In one article, according to records of the National Health Insurance Corporation of Korea, there were 7,937,000 levosulpiride prescriptions in Korean during 2007, and this number was approximately double the 4,318,400 metoclopramide prescriptions. This systematic process(NHIS) ensures that the registration data is reliable And 237 were excluded because they did not match the control group.[6]

9) Page 3, Line 88 – are these newly prescribed or any fill? Please clarify.
[Answer] Thank you for your comments. We edited and clarify in detail in Page 3 method.

10) Page 4, Line 115 – shouldn’t it be 1-20 day group and not 0-20 group since ‘none’ is separate in the analysis on Table 2 (this would be the same for levosulpriride)

[Answer] Thank you for your comments. Your opinion is correct. We are sorry for the unintended mistake groups seperation. We edited 3 groups seperation in all tables(none, 1-19days, 20 days).

Discussion

11) Page 5, Lines 144-152 – this is doesn’t appear to add to the study since it is discussing bench science and not the clinical implications.

[Answer] Thank you for your comments. As you recommend, we deleted that.

12) Page 5, Line 153-158 – this seems to be more relevant in the introduction than the discussion.

[Answer] Thank you for your comments. As you recommended, these were moved to an appropriate location in the ‘Introduction’ section.

13) Page 5, Line 159 – noted ‘many studies have demonstrated that antiemetics are associated with DIP” but does not cite – please cite these many studies To note, the previous study evaluating this prescribing cascade was not cite in this manuscript (PMID: 7500509) which should both be mentioned in the intro and here. Especially since your results are very similar to the results by Avorn et al (and that the number of exposure days being dose dependent is new to your study).

[Answer] Thank you for your comments. As you recommend, the phrase was edited as follows: ‘Many studies have already demonstrated that antiemetics are associated with DIP.[6,7] In one study in united states, metoclopramide users were three times more likely to start using levodopa-containing drugs than nonusers (OR = 3.09; 95% confidence interval [CI], 2.25-4.26). In addition, increased risk with increasing daily metoclopramide dose: OR is 1.19 (95% CI, 0.50 to 2.81) above 0-10 mg per day, 3.33 (95% CI, 1.98 to 5.58) above 10-20 mg per day, It was 5.25 (95% CI, 1.16-8.50) more than 20 mg a day.[6]

14) Page 5, Line 168 – “As other studies”, please cite and described these other studies.

[Answer] Thank you for your comments. As you recommend, the phrase was edited as follows : ‘As other studies have shown that DIP is dose-dependent, the subsequent prescription of levodopa in this study has also increased with longer use of gastroprokinetics.[7,8] In one study in south korea, the ORs tended to be higher in those exposed to higher chronic cumulative doses, had a significant association with the increased risk of parkinsonism, depending on cumulative dose.[7]

In our study, the longer the exposure period, it was found that severe with levosulpiride than metoclopramide.

15) Page 6, Line 175 – “Not all cases of this paper can be called DIP” – I’m not sure what you mean. Nonetheless, I think an explanation is needed.

[Answer] Thank you for your comments. The phrase was edited and clarify as follows:

‘First, not all cases of this paper can be called DIP. Because this study did not consider NHIS Parkinson's disease diagnostic code and analyzed people who were prescribed levodopa medication, there is a limitation that those who were prescribed levodopa may not necessarily be diagnosed with DIP.’

16) Page 6, 175 – “PD medications other than levodopa were not considered” Need to explain why not (levodopa is the primarily first line agent for parkinsonism?)

[Answer] Thank you for your comments. The phrase was edited and clarify as follows:

‘Second, antiparkinsonian medications other than levodopa were not considered. Because the DIP treatment does not necessarily use levodopa, there are limitations that other antiparkinsonian disease treatments may also be used.’

17) Page 6, line 180 – I’m having trouble connecting your explanation to selection bias – isn’t it applicable to both treatment and control?

[Answer] Thank you for your comments. The phrase was edited as follows:

‘Third, this study is based on NHIS-Senior Cohort Database, which it consisted of elderly adults with national health checkup results. Therefore, there may be a selection bias because it is more likely to include people who are more concerned about their health than those who did not receive regular medical checkup.’

Conclusion

18) Typically, I don’t see citations in the conclusion I don’t think discussing the steps to prevent prescribing cascade is relevant part of your conclusion At no point do you mention metoclopramide, levosulpiride, or levodopa in your conclusion – focus on the conclusion of you results and not general prescribing cascades.

[Answer] Thank you for your comments. As you recommend, we focus on the conclusion of preceding metoclopramide or levosulpiride use and subsequent levodopa prescription. The phrase was edited as follows: ‘Metoclopramide and levosulpiride were frequently prescribed within 90 days of a prescription for levodopa. Before prescribing levodopa, we recommend to consider whether it is a side effect caused by metoclopramide and levosulpiride.’

Reference

Petrone, K.; Katz, P. Approaches to appropriate drug prescribing for the older adult. Primary care 2005, 32, 755-775, doi:10.1016/j.pop.2005.06.011. Vouri, S.M.; van Tuyl, J.S.; Olsen, M.A.; Xian, H.; Schootman, M. An evaluation of a potential calcium channel blocker-lower-extremity edema-loop diuretic prescribing cascade. Journal of the American Pharmacists Association : JAPhA 2018, 58, 534-539 e534, doi:10.1016/j.japh.2018.06.014. Ma, H.I.; Kim, J.H.; Chu, M.K.; Oh, M.S.; Yu, K.H.; Kim, J.; Hahm, W.; Kim, Y.J.; Lee, B.C. Diabetes mellitus and drug-induced Parkinsonism: a case-control study. Journal of the neurological sciences 2009, 284, 140-143, doi:10.1016/j.jns.2009.05.006. Lopez-Sendon, J.L.; Mena, M.A.; de Yebenes, J.G. Drug-induced parkinsonism in the elderly: incidence, management and prevention. Drugs & aging 2012, 29, 105-118, doi:10.2165/11598540-000000000-00000. Shin, H.W.; Chung, S.J. Drug-induced parkinsonism. Journal of clinical neurology (Seoul, Korea) 2012, 8, 15-21, doi:10.3988/jcn.2012.8.1.15. Shin, H.W.; Kim, M.J.; Kim, J.S.; Lee, M.C.; Chung, S.J. Levosulpiride-induced movement disorders. Movement disorders : official journal of the Movement Disorder Society 2009, 24, 2249-2253, doi:10.1002/mds.22805. Kim, S.; Cheon, S.M.; Suh, H.S. Association Between Drug Exposure and Occurrence of Parkinsonism in Korea: A Population-Based Case-Control Study. The Annals of pharmacotherapy 2019, 10.1177/1060028019859543, 1060028019859543, doi:10.1177/1060028019859543. Avorn, J.; Gurwitz, J.H.; Bohn, R.L.; Mogun, H.; Monane, M.; Walker, A. Increased incidence of levodopa therapy following metoclopramide use. Jama 1995, 274, 1780-1782.

Reviewer 2 Report

Review of Manuscript

            Current study addresses an important topic - Drug-induced Parkinsonism (DIP) in elderly patients in a large cohort of population in South Korea.  It is well designed, well written and empowered with good statistical analyses.  The sample size is large and has been well addressed for confounding factors such as age, sex and use of other psychotic drugs.  Overarching goal of the study is to examine the prescribing cascades underlying DIP.Drug-induced parkinsonism is a common etiology of parkinsonism and is more critical in the elderly patients. The study delineates the association between the initiation of L-DOPA therapy and the use of two antiemetic and/or gastroprokinetic drugs metoclopramide and levosulpride. Although these drugs are administered based on their ability to dopamine2-receptor blockade in the area postrema, they also antagonize the central D2 receptors, leading to adverse effects including extrapyramidal side effects, and thus are capable of inducing DIP.

A major advantage of the study is that the drug levosulpride is only used in S Korea and Italy, hence this study provides valuable data on this drug.  Prolonged use of both of these drugs was found to be more detrimental; Levosulpride was found to be more harmful between the two drugs.  

A minor concern:

The font-size in the Figure 1 (The Flow Chart) is not clearly legible and overall the image is blurry.

Author Response

[Point by point response to the Reviewer’s comments]

Reviewer #2’s comments:

1) The font-size in the Figure 1 (The Flow Chart) is not clearly legible and overall the image is blurry.

[Answer] Thank you for your comments. The font size and details was modified to give a clear view of the image.

Figure 1. Flowchart of Study Subjects

Round 2

Reviewer 1 Report

I feel that this paper will add nicely to the literature after the updates the authors made. 

There are still a few minor details that need to be updated prior to publication.

1) There are still many grammar and spelling issues. United States and South Korea were not capitalized. Levodopa was spelled as levopa in a few cases. Missing words like 'of' and 'the'. I recommend a review of grammar and spelling by the journal or another outside person.

2) Formatting in Figure 1 was difficult to read. I would recommend bolding key words, using (n=XXXX) to describe the populations, do not double space, and think 'left align' may look clearer than 'center' within the flow chart.

Beyond that, I have no further comments.

Author Response

[Point by point response to the Reviewer’s comments]

Reviewer #1’s comments:

1) There are still many grammar and spelling issues. United States and South Korea were not capitalized. Levodopa was spelled as levopa in a few cases. Missing words like 'of' and 'the'. I recommend a review of grammar and spelling by the journal or another outside person.

[Answer] Thank you for your comments. We have corrected all words that were incorrectly spelled levopa as levodopa. And we received an English calibration as you recommended.

2) Formatting in Figure 1 was difficult to read. I would recommend bolding key words, using (n=XXXX) to describe the populations, do not double space, and think 'left align' may look clearer than 'center' within the flow chart.

[Answer] Thank you for your comments. As you recommended, we modified flow chart as follows:
